# Mapping Land Use/Land Cover Changes and Forest Disturbances in Vietnam Using a Landsat Temporal Segmentation Algorithm

**Katsuto Shimizu** [1,*][ID]**, Wataru Murakami** [2]**, Takahisa Furuichi** [2] **and Ronald C. Estoque** [3][ID]

1. Department of Forest Management, Forestry and Forest Products Research Institute, 1 Matsunosato, Tsukuba 305-8687, Ibaraki, Japan
2. Department of Disaster Prevention, Meteorology and Hydrology, Forestry and Forest Products Research Institute, 1 Matsunosato, Tsukuba 305-8687, Ibaraki, Japan
3. Center for Biodiversity and Climate Change, Forestry and Forest Products Research Institute, 1 Matsunosato, Tsukuba 305-8687, Ibaraki, Japan
* Correspondence: katsutoshimizu@ffpri.affrc.go.jp; Tel.: +81-29-829-8314

**Abstract:** Accurately mapping land use/land cover changes (LULCC) and forest disturbances provides valuable information for understanding the influence of anthropogenic activities on the environment at regional and global scales. Many approaches using satellite remote sensing data have been proposed for characterizing these long-term changes. However, a spatially and temporally consistent mapping of both LULCC and forest disturbances at medium spatial resolution is still limited despite their critical contributions to the carbon cycle. In this study, we examined the applicability of Landsat time series temporal segmentation and random forest classifiers to mapping LULCC and forest disturbances in Vietnam. We used the LandTrendr temporal segmentation algorithm to derive key features of land use/land cover transitions and forest disturbances from annual Landsat time series data. We developed separate random forest models for classifying land use/land cover and detecting forest disturbances at each segment and then derived LULCC and forest disturbances that coincided with each other during the period of 1988–2019. The results showed that both LULCC classification and forest disturbance detection achieved low accuracy in several classes (e.g., producer's and user's accuracies of 23.7% and 78.8%, respectively, for forest disturbance class); however, the level of accuracy was comparable to that of existing datasets using the same reference samples in the study area. We found relatively high confusion between several land use/land cover classes (e.g., grass/shrub, forest, and cropland) that can explain the lower overall accuracies of 67.6% and 68.4% in 1988 and 2019, respectively. The mapping of forest disturbances and LULCC suggested that most forest disturbances were followed by forest recovery, not by transitions to other land use/land cover classes. The landscape complexity and ephemeral forest disturbances contributed to the lower classification and detection accuracies in this study area. Nevertheless, temporal segmentation and derived features from LandTrendr were useful for the consistent mapping of LULCC and forest disturbances. We recommend that future studies focus on improving the accuracy of forest disturbance detection, especially in areas with subtle landscape changes, as well as land use/land cover classification in ambiguous and complex landscapes. Using more training samples and effective variables would potentially improve the classification and detection accuracies.

**Keywords:** LULCC; Landsat; LandTrendr; disturbance; Google Earth Engine; random forest

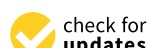



## 1. Introduction

Land use/land cover changes (LULCC), especially in forest areas, can lead to carbon emissions, result in biodiversity loss, and exacerbate the risk of natural hazards at both the regional and global scales [1–3]. In addition, forest disturbances, which include changes that do not cause LULCC (e.g., temporary loss of forest canopy, forest degradation, and

harvesting followed by replanting), can also affect carbon sequestration and influence the susceptibility of areas or regions to natural hazards [4]. Regular mapping of LULCC and forest disturbances by using remote sensing data provides information for understanding the influence of anthropogenic activities on the environment [5,6]. In the context of climate change, carbon emissions related to LULCC have large uncertainties, which has yielded conflicting results in global carbon studies [7]. Therefore, accurate mapping of the spatial and temporal patterns of LULCC and forest disturbances is needed for effectively monitoring carbon emissions and timely decision-making.

Satellite remote sensing is an efficient tool for monitoring across a large area because of its cost-effectiveness, wide spatial coverage, and frequent observation intervals [8,9]. Various approaches for mapping land use/land cover (LULC) and forest disturbances have been proposed using different satellite data (e.g., optical and radar) at varying spatial and temporal resolutions, extents, and periods. Mapping efforts have led to numerous annual to decadal LULC/forest disturbance products at both the regional and global scales [10]. For example, Dynamic World is a near real-time high-spatial resolution LULC product based on Sentinel-2 data and a deep learning architecture that covers the global land surface [11]. Other examples are ESA WorldCover, a global 10 m LULC dataset for 2020 that is generated by the European Space Agency (ESA) using Sentinel-1 and Sentinel-2 data [12] and Copernicus Global Land Cover at 100 m for 2015–2019 [13]. The accuracies and information obtained in these maps are unique to the products; therefore, users are required to select appropriate approaches and satellite data to achieve their specific objectives. Especially for long-term mapping, which can span more than three decades, global products are usually not available due to a lack of data for the past. Thus, locally adjusted products need to be generated.

With significant development in satellite remote sensing algorithms, numerous approaches have been proposed for LULC mapping and forest disturbance detection, especially using Landsat data [14]. Change detection algorithms including the breaks for additive seasonal and trend (BFAST) [15,16], Landsat-based detection of trends in disturbance and recovery (LandTrendr) [17], and continuous change detection and classification (CCDC) [18] have been proposed and applied in many regions (e.g., [19–23]). In particular, the LandTrendr temporal segmentation is a widely used algorithm for detecting forest disturbances and recovery that enhances an annual Landsat time series spectral index to generate temporal segments and eliminate spectral noise by fitting a sequence of straight lines [17]. LandTrendr is easy to implement in large areas for detecting both abrupt and gradual changes, and it is less computationally intensive compared to other algorithms that use all available Landsat data, such as CCDC [24]. Because of these advantages, LandTrendr has been extensively applied to various forest environments [25,26] and further improved by introducing ensemble learning for change detection. Cohen et al. [27] applied a secondary classification approach using random forest (RF) [28] for segmentation in LandTrendr rather than depending on a fixed threshold for identifying forest disturbances as in the original algorithm. Similarly, Nguyen et al. [29] used RF to classify disturbances and recovery based on LandTrendr temporal segmentation. The temporal segments of LandTrendr represent critical features of spectral trajectories [17] and can be used to extract key information to characterize changes in land surface (e.g., [30,31]). Such characteristics of the temporal segmentation are potentially useful for LULCC, but the applicability of such an approach has not been investigated in detail. More specifically, there are limited studies that investigated effective approaches for handling a large number of features derived from temporal segmentation for LULCC, which generally shows more diverse spectral changes in time series than forest disturbances. In this regard, the use of machine learning algorithms is suited to predict LULC/LULCC. Numerous studies have implemented LULC/LULCC classification using machine learning algorithms such as RF, support vector machines, and deep neural networks (e.g., [32–35]). These previous studies, however, mostly focused on limited time periods and did not map long-term changes. Landsat time series analysis is potentially utilized for consistent mapping of long-term

LULCC that is consistent with forest disturbance at the same time. This mapping approach provides better insights into land surface dynamics.

The objective of this study was to investigate the applicability of a Landsat time series segmentation algorithm to map the annual LULCC and forest disturbances for a long-term period of more than 30 years. We selected the northern part of Vietnam as the study area and developed RF models using LandTrendr temporal segmentation. Then, we compared the estimation accuracies of the RF models to investigate the utility of Landsat time series analysis for mapping LULCC and forest disturbances. Vietnam is a densely populated country in Southeast Asia, experiencing a high degree of forest loss and reforestation in the past several decades. In Vietnam, forest areas decreased due to human pressure until the 1980s; however, reforestation activities increased the forest cover since the 1990s [36]. At the same time, Vietnam experienced significant urbanization in recent years [37]. Rapid LULCC in the last several decades can exacerbate the risk of natural hazards, such as flooding and landslides, especially in mountainous areas [38,39]. The increased natural hazards in the mountainous regions of Vietnam have caused human losses, economic damage, and abandonment of agricultural land in the rural environment [40,41] Thus, there is an urgent need to map LULCC and forest disturbances and to understand their influences in Vietnam. Recently, optical and radar satellite data have been used to map LULCC (e.g., [42–44]) and forest disturbances (e.g., [45]) in Vietnam and other countries in Southeast Asia (e.g., [46–48]). However, the long-term trends of LULCC and forest disturbances are still unclear. To understand the causes and consequences of LULCC and forest disturbances, a mapping approach that utilizes frequently acquired long-term satellite observations is needed.

## 2. Methods

### 2.1. Study Area

Figure 1 shows the study area, which is the part of Vietnam north of 20°N and covers an area of about 12.2 million ha. The topography is characterized by steep terrain in the northwest inland area and relatively flat terrain in the south eastern coastal area at the Red River Delta. According to Köppen's classification, the climate is warm temperate with a hot summer and a relatively cold season in the mountainous regions. The rainy season normally starts in April or May and lasts until October. The forests in mountainous and flat areas are dominated by broadleaf and tropical deciduous species, respectively.

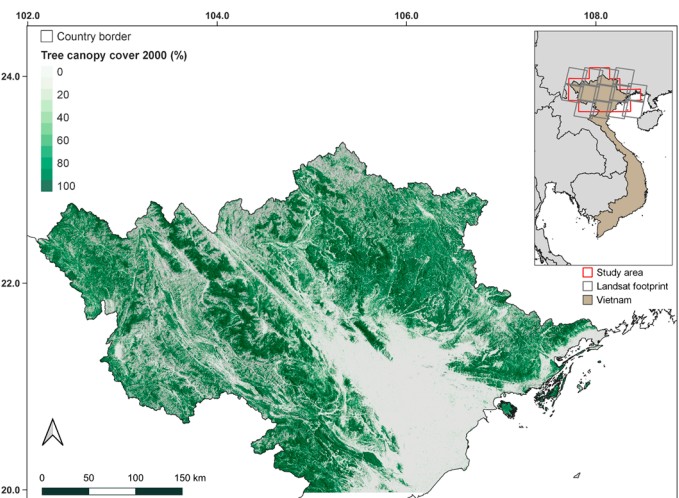

**Figure 1.** Study area in northern Vietnam. The tree canopy cover in 2000 from Hansen et al. [49] was overlaid on a digital elevation model from the Shuttle Radar Topography Mission [50]. The country boundary dataset was sourced from Global Administrative Areas, v3.4 [51].

According to the Food and Agriculture Organization (FAO) [52], 47% of Vietnam was covered by forest in 2020. Despite the forest recovery over the past three decades, there exist regional variations in deforestation, forest degradation, and recovery [53]. Currently, plantation forests comprising mainly of acacia, eucalyptus, and rubber account for about 30% of the forest area. Extensive anthropogenic activities have created a mosaic landscape of primary forests, secondary forests, plantation forests, shrubs, and agricultural land in the northern upland areas [53]. The agriculture and urban areas are distributed over the flat coastal region.

### 2.2. Processing Flow

Figure 2 shows the processing flow of the Landsat time series segmentation and RF modeling for mapping LULCC and forest disturbances. We first implemented LandTrendr temporal segmentation [17] by using the annual Landsat time series data in Google Earth Engine [54]. We collected training data for tuning RF models. Then, we developed separate RF models for LULC classification and forest disturbance detection based on temporal segmentation results and collected training data. After LULC and forest disturbances were predicted for each segment of the study area, post-processing procedures were applied to correct obviously misclassified LULCC and forest disturbances. We used the final classification results of each temporal segment to map the annual LULC/LULCC and forest disturbances for the entire study area.

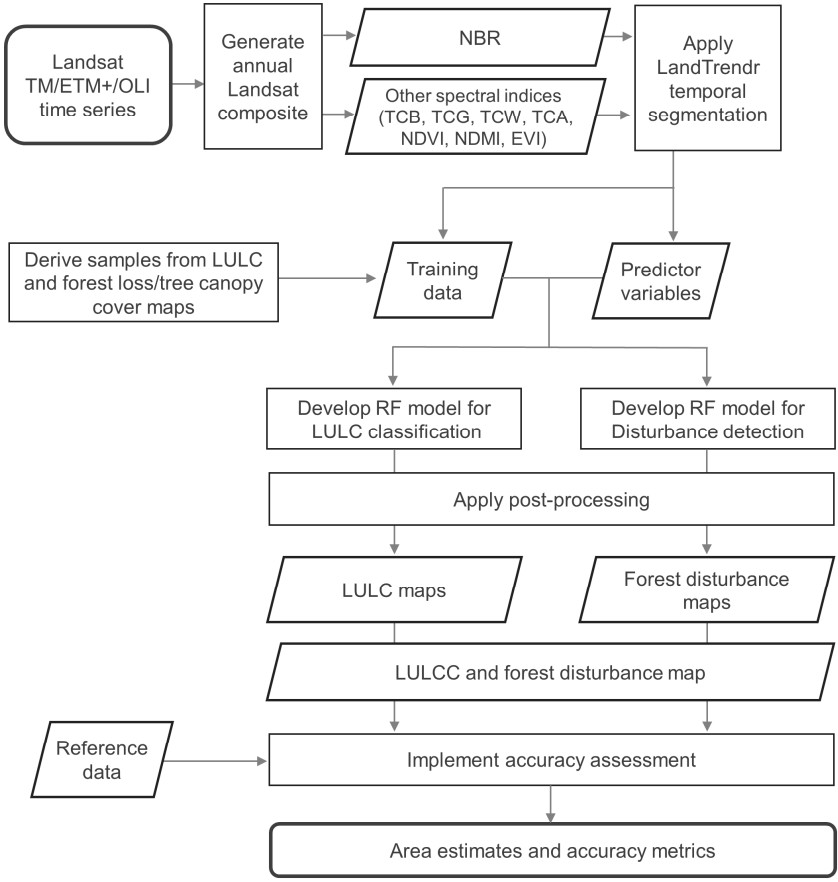

**Figure 2.** Processing flow for mapping land use/land cover changes (LULCC) and forest disturbances.

### 2.3. Landsat Time Series Data

The primary satellite data for time series analysis in this study was annual composites of Landsat data. We used the archive of Landsat TM/ETM+/OLI Collection 1 Tier 1 surface reflectance (SR) data [55,56] from 1987 to 2021 in Google Earth Engine (GEE). We determined the duration according to whether sufficient Landsat data were available to generate

annual composites for the study area. To reduce the yearly variations caused by seasonal changes, we only used images collected between 1 May and 30 November. We determined this period by considering the green up and down dates in the growing cycle of forests in the study area from the MODIS land cover dynamics product (MCD12Q2 [57]). We calibrated the OLI SR data to TM/ETM+ data by using the regression coefficients provided by Roy et al. [58]. We removed the pixels affected by clouds, cloud shadows, and snow by using the quality assessment (QA) bands from CFmask [59,60]. To generate annual Landsat image composites from the collection of Landsat data, we employed medoid compositing using the LT-GEE code (https://emapr.github.io/LT-GEE/index.html, accessed on 21 December 2021). We performed the image compositing for each year using any Landsat sensor. The processing resulted in a single SR composite image for each year from 1987 to 2021.

### 2.4. LandTrendr Temporal Segmentation

We implemented the LandTrendr algorithm for temporal segmentation [17,61] in GEE. LandTrendr fits straight-line temporal segments to the trajectory of a spectral index in a pixel time series. We calculated the normalized burn ratio (NBR) [62] as a spectral index to determine the temporal segments of LandTrendr owing to its suitability for characterizing forest dynamics [63]. To run LandTrendr, a set of parameters is required to identify breakpoints and fit straight lines. We used the same default values for parameters as Kennedy et al. [61] except for maxSegments (set to 8), recoveryThreshold (set to 1), and pvalThreshold (set to 0.1, Table S1 in Supplementary Materials). These modifications were made to capture multiple changes and rapid vegetation recovery based on the findings of previous studies [29,64–68].

Additionally, we also derived other spectral indices, namely the tasseled cap brightness (TCB), greenness (TCG), wetness (TCW) [69], and angle (TCA) [70]; enhanced vegetation index (EVI) [71], normalized difference vegetation index (NDVI) [72,73], and normalized difference moisture index (NDMI) [74] composites. We then applied them to the fit-to-vertex (FTV) procedure [75], which forces spectral indices fit the timing of breakpoints determined by NBR segmentation. As a result of the FTV procedure, each pixel had the fitted straight-line trajectories of NBR and the other seven spectral indices, which had the same timing and duration segments but different spectral properties. We included the FTV procedure because information from different spectral indices for each segment is useful for describing LULCC and forest disturbances [31,75].

### 2.5. RF Models for LULC Classification and Disturbance Detection

After the temporal segmentation with LandTrendr, we derived a set of predictor variables from each segment (Table S2). The two RF models for LULC and disturbance detection shared the same predictor variables. For NBR, we computed the start value, end value, spectral magnitude, duration, disturbance signal-to-noise ratio (DNSR) [27], change rate, and relative change of each segment. In addition, the fitted NBR was used to calculate the start value, spectral magnitude, duration, DNSR, and change rate of the pre- and post- segments as predictor variables. For the fitted spectral indices from FTV, we computed the start value, spectral magnitude, change rate, and relative change. In total, we extracted 45 predictor variables from each temporal segment. The complete list of the predictor variables is shown in Table S2.

We collected training data for the RF models of LULC classification and disturbance detection by visual interpretation of high-spatial-resolution images in Google Earth and Landsat time series data (Table 1). In this study, we defined a forest disturbance as any discrete event that causes a reduction of forest canopy visible from high-spatial-resolution satellite images. We used tree canopy cover for 2000 and forest loss maps for 2001–2019 from Hansen global forest change (GFC) data [49] and a LULC map of Vietnam in 2019 [42] to sample random locations. First, we used the tree canopy cover and forest loss maps to generate a map consisting of forest (tree canopy cover $\geq$ 10%) without forest loss, non-

forest (tree canopy cover < 10%), forest loss, and forest without forest loss but within 1 pixel from forest loss pixels. Then, we randomly selected pixel locations from each class of the map. The temporal segment was the primary unit for the RF models in this study. Thus, we allocated reference labels (both LULC classes and forest disturbance/no-disturbance) for each temporal segment at the sampling pixel locations, as shown in Figure 3. After the predictor variables were derived from the temporal segments, we developed preliminary RF models that we applied to the entire study area. The resultant prediction was used to generate LULC map for 2019, which we compared against the LULC map from Phan et al. [42] after adjusting the LULC classes. Then, we collected training data for locations where the LULC classes disagreed. We iteratively collected training samples by using updated preliminary maps and the LULC disagreement. Based on the results, we finally collected training samples comprising 9592 segments at 2210 pixel locations.

**Table 1.** Land use/land cover (LULC) classes and their descriptions.

| Class | Description |
|---|---|
| Cropland | Agricultural land such as paddy fields and cultivated areas |
| Barren | Bare soil without vegetation cover or sparse shrub vegetation |
| Forest | Areas with a tree canopy cover of >10% and height potentially taller than 5 m, including secondary forests and plantation forests |
| Grass/Shrub | Grassland and woody vegetation that is not forest |
| Settlement | Residential and built-up areas including unpaved roads |
| Water | Water bodies including rivers, lakes, ponds, inundations, and sea |

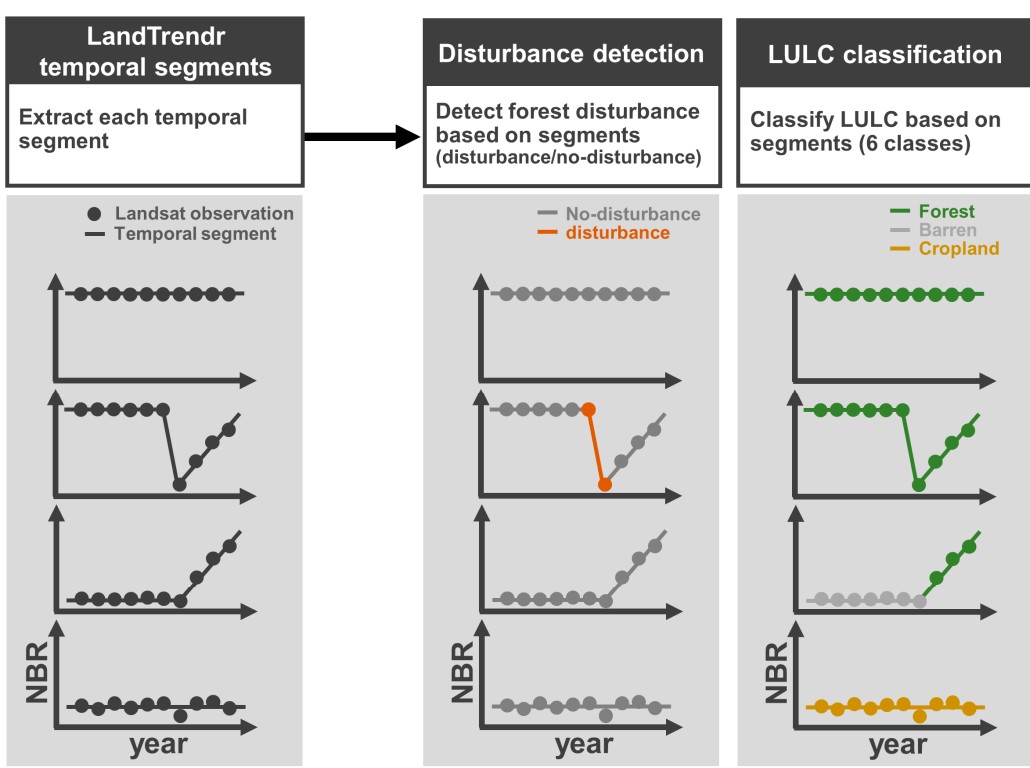

**Figure 3.** Illustration of the LULC classification and forest disturbance detection based on the LandTrendr temporal segmentation.

We used the collected training data to train the RF models (i.e., smileRandomForest) for LULC classification and forest disturbance detection in GEE. We set the number of variables per split as the square root of the number of predictors with a number of trees of 500. We exported the segments classified by the RF models and applied post-processing procedures to each segment to correct obvious misclassifications or inconsistent predictions in

R version 4.1.1 [76]. During the post-processing, we applied a 3 × 3 pixel spatial majority class filter to the mapped LULC of the first segments. Because forest disturbances that occurred in non-forest areas are not logical, we replaced such detections with no-disturbance. For LULCC, LULC transitions from forest to other classes without forest disturbance were eliminated. Finally, we removed forest disturbances of less than six spatially adjacent pixels to suppress false positives. After the post-processing, we derived annual LULC and forest disturbance maps. Although the RF prediction of LULC and disturbance detection covered a slightly longer period, we only used the mapping results for 1988–2019 because the LandTrendr-like algorithm (i.e., offline change detection, [26]) sometimes causes erroneous detection at the end of the time series [77].

We used the LULC 1988, LULC 2019, and forest disturbance maps to generate a combined LULCC and forest disturbance map for 1988–2019. Because we would like to map LULCC for this period and avoid the emergence of too many LULCC classes, we only considered LULC in 1988 and 2019 and ignored possible multiple LULCC for a particular location during this period. In addition, we merged several LULCC classes because of the large numbers of "from-to" classes. The LULCC and forest disturbance map had 11 classes comprising six classes representing stable LULC during 1988–2019, two classes representing consolidated LULCC, and three classes representing forest disturbance. These classes are shown in Table S3.

*2.6. Accuracy Assessment*

We assessed the accuracies of the maps for LULC 1988, LULC 2019, forest disturbance, and combined LULCC and forest disturbance for 1988–2019. We assessed the accuracies of these maps based on the corresponding map classes as explained in the following paragraphs. Although most previous studies removed non-forest areas when assessing the accuracy of forest disturbance class (e.g., [78,79]), we included the entire study area (i.e., both forest and non-forest classes) to consider possible omission errors that occurred while classifying the non-forest areas, which were actually forest areas. We followed the recommendation by Olofsson et al. [80] for sampling design, response design, and analysis protocols for assessing the accuracy and estimating area of mapped classes. We collected reference samples based on a stratified random sampling. We generated strata by using the mapped LULC and forest disturbances (i.e., 11 classes in Tables S3 and S4) with an additional spatial buffer stratum, as given in Table 2. As suggested by Olofsson et al. [81], a spatial buffer stratum is a simple and useful way to reduce the uncertainty of the producer's accuracy (PA) and area of forest disturbance classes that occupy a smaller area than a stable forest (i.e., forest without experiencing forest disturbance) stratum when stratified random samples are used. Based on the insight that omission errors of forest disturbances are likely to occur in proximity to detected disturbance pixels, previous studies have utilized a spatial buffer stratum that surrounds disturbance pixels with different buffer sizes (e.g., [20,82–85]). In this study, we assigned pixels of stable forest class within 1 pixel from forest disturbance pixels as a buffer stratum where the buffer size was determined based on the area weights of the disturbance (7.4%) and stable forest classes (60.7%). Note that we generated the strata by using interim mapping results, not the final version. This is because the study was a part of a project, and accuracy assessment was required at the development stage. Thus, we carefully implemented the sample collection to not violate the independence of the reference samples for the assessment of the final maps [85]. This sampling design did not affect the unbiasedness of the estimators. We determined the sample size for accuracy assessment as follows [80,86]:

$$n = \left( \frac{\sum W_i S_i}{S(\hat{O})} \right)^2 \tag{1}$$

$$S_i = \sqrt{U_i(1 - U_i)} \tag{2}$$

where $n$ is the sample size, $W_i$ is the area weight of class $i$, $S_i$ is the standard deviation of stratum $i$, $S(\hat{O})$ is the standard error of the assumed overall accuracy (OA), and $U_i$ is the assumed user's accuracy (UA) of stratum $i$. We assumed UA as 80% for forest disturbance classes and 75% for the rest of the classes after trying several potential values, as suggested by Stehman and Foody [87]. With the 95% confidence interval of OA as 3%, we obtained a sample size of 792 for the accuracy assessment. We used 30 m pixels as the spatial assessment unit for accuracy assessment. The sample allocation to the strata was determined following the allocation procedure suggested by Olofsson et al. [80] with at least 50 samples for each strata (Table 2).

**Table 2.** Sample allocation to each stratum in accuracy assessment.

| Class | Initial Labeling (%) | Reinterpretation (%) | Final (%) |
|---|---|---|---|
| LULC 1988 | 65.3 | 93.2 | 100 |
| LULC 2019 | 81.6 | 94.8 | 100 |
| Forest disturbance | 80.4 | 92.3 | 100 |

We visually interpreted each reference sample by assigning reference classes. Two independent interpreters labeled the LULC classes as defined in Table 1 for 1988 and 2019 and the occurrence of forest disturbance within the period of 1988–2019 with the aid of the Collect Earth software developed by FAO [88]. We prepared a time series NDVI and normalized difference fraction index (NDFI) [89] from all available Landsat data for 1987–2021 with a CCDC harmonic regression model fitting [83] in Collect Earth. Prior to the interpretation of the reference samples, we randomly collected 50 samples independent of the reference samples and used them for training the two interpreters, along with a class-labeling manual that describes typical spectral characteristics of each LULC class. Then, the two interpreters independently determined the reference labels of the 792 reference samples with their confidence in the interpretation (i.e., initial labeling). The interpreters did not know the stratum for each reference sample while implementing visual interpretation. After the initial labeling, inconsistent reference samples that had a disagreement in either LULC 1988, LULC 2019, or disturbance labels were identified and reinterpreted by both interpreters on whether to change the labels. For the remaining inconsistent samples after the reinterpretation, the interpreters visually checked the samples together and determined the reference labels after discussion with a field specialist to allow for correct label assignment (Table 3). The interpreters had difficulty determining the reference labels for some of the reference samples for LULC 1988 because of the lack of high-spatial-resolution data. Thus, the interpretation confidence was also used to determine the label in such cases. The reference labels of all the reference samples from the two interpreters were matched by these procedures (Table 3). Because we had the four maps for accuracy assessment, we converted the labels of the reference samples to be in accordance with the classes of each map when assessing accuracies and estimating areas. The classification agreement was defined as the match between the map and reference classes. Additionally, we manually delineated spatially contiguous forest disturbance patches that intersected with labeled disturbance in the reference samples by using annual Landsat RGB composites and rasterized them to calculate disturbance patch size. We conducted this manual delineation to calculate the PA of disturbance for specific disturbance size classes. This procedure only added the disturbance patch size information to the pixel-based reference samples and did not affect the classification agreement.

**Table 3.** Proportion of label agreement between two interpreters at each stage of visual interpretation.

| Type | Stratum | Area Weight | Sample Size |
|---|---|---|---|
| Stable LULC | Cropland | 0.134 | 61 |
| | Barren | 0.000 | 50 |
| | Forest | 0.504 | 231 |
| | Grass/Shrub | 0.023 | 50 |
| | Settlement | 0.021 | 50 |
| | Water | 0.016 | 50 |
| LULCC | Others to Forest | 0.080 | 50 |
| | Others to Others (excluding forest) | 0.046 | 50 |
| Forest disturbance | Disturbance with forest to forest | 0.071 | 50 |
| | Disturbance with forest to others | 0.002 | 50 |
| | Disturbance with others to forest | 0.001 | 50 |
| Buffer | Buffer on stable forest | 0.103 | 50 |

We calculated the PA and UA of each class and OA with the estimated population error matrices [90,91]. As we collected reference samples by using the strata from the interim maps, the final maps and sampling strata inevitably differed in this study. Therefore, we used the indicator functions and combined ratio estimator proposed by Stehman [91] to estimate accuracies and area. The OA and area of each class were estimated as follows:

$$\hat{Y} = \frac{1}{N} \sum_{h=1}^{H} N_h \overline{p}_h \tag{3}$$

where $N$ is the total number of pixels in the population, $H$ is the number of strata, $N_h$ is the total number of pixels in stratum $h$, and $\overline{p}_h$ is the sample means of correctly classified pixels (for OA) or the sample proportions of the specific reference class (for area) in stratum $h$ as defined in the indicator functions. The variance estimator is given by Stehman [91]:

$$\hat{V}(\hat{Y}) = \frac{1}{N^2} \sum_{h=1}^{H} N_h{}^2 (1 - n_h/N_h) s_{yh}^2 / n_h \tag{4}$$

where $n_h$ is the number of sample pixels in stratum $h$ and $s_{yh}^2$ is the sample variances for $y_u$ in stratum $h$. The estimates of the PA and UA of each class were calculated as follows [91]:

$$\hat{R} = \frac{\sum_{h=1}^{H} N_h \overline{y}_h}{\sum_{h=1}^{H} N_h \overline{x}_h} \tag{5}$$

where $\overline{y}_h$ and $\overline{x}_h$ are the sample means of $y_u$ and $x_u$, respectively, in stratum $h$ and $y_u$ and $x_u$ are the defined indicator functions for each accuracy metric using pixel $u$. We obtained the variance of $\hat{R}$ using the following formulas by Stehman [91]:

$$\hat{V}(\hat{R}) = \left(\frac{1}{\hat{X}^2}\right) \left[ \sum_{h=1}^{H} N_h^2 \left(1 - \frac{n_h}{N_h}\right) \left(s_{yh}^2 + \hat{R}^2 s_{xh}^2 - 2\hat{R} s_{xyh}\right) / n_h \right] \tag{6}$$

where $s_{xh}^2$ is the sample variance for $x_u$ in stratum $h$. $\hat{X}$ and $s_{xyh}$ are defined as follows [91]:

$$\hat{X} = \sum_{h=1}^{H} N_h \overline{x}_h \tag{7}$$

$$s_{xyh} = \sum_{u=1}^{n_h} (y_u - \overline{y}_h)(x_u - \overline{x}_h) / (n_h - 1) \tag{8}$$

We applied the estimator formulas to the maps of LULC 1988, LULC 2019, forest disturbance, and combined LULCC/disturbance. We obtained 95% confidence intervals for each estimate. For the PA of disturbance detection, we also calculated the PA for disturbance sizes of <1 ha and ≥1 ha to investigate the performance of detecting small forest disturbances using reference disturbance size information collected through visual

interpretation. For performance comparison with existing disturbance detection and LULC classification datasets, the accuracy metrics were calculated for the forest loss map of Hansen GFC (2000–2019) and LULC map of 2019 for Vietnam of Phan et al. [42] using the same reference samples and estimators after clipping these maps to our study area.

## 3. Results

### 3.1. Accuracy Assessment

The results revealed that the OAs of the LULC classification for 1988 and 2019 were 67.6% (±3.9% in the 95% confidence interval) and 68.4% (±3.8%), respectively (Table 4). The population error matrices revealed large omission errors for the grass/shrub and forest classes in both LULC classifications (Tables S5 and S6). The classification errors between the cropland and grass/shrub classes were also large. Both PA and UA were generally high for the stable forest class. PA and UA. On the other hand, relatively low accuracies were achieved for the barren and grass/shrub classes for both LULC 1988 and 2019.

**Table 4.** Accuracy assessment for the 1988 and 2019 classified LULC maps. The producer's accuracy (PA) and user's accuracy (UA) for each class and overall accuracy (OA) are shown with 95% confidence intervals.

| Class | LULC 1988 | | | LULC 2019 | | |
|---|---|---|---|---|---|---|
| | PA (%) | UA (%) | OA (%) | PA (%) | UA (%) | OA (%) |
| Cropland | 59.1 (±6.5) | 75.7 (±7.6) | 67.6 (±3.9) | 49.2 (±7.2) | 69.4 (±10.4) | 68.4 (±3.8) |
| Barren | 0.0 (±0.0) | 25.0 (±12.8) | | 0.2 (±0.4) | 4.4 (±10.0) | |
| Forest | 96.4 (±2.0) | 69.7 (±5.0) | | 96.7 (±1.8) | 69.6 (±4.5) | |
| Grass/Shrub | 11.8 (±5.4) | 33.3 (±13.3) | | 9.2 (±4.9) | 35.1 (±15.3) | |
| Settlement | 41.1 (±20.3) | 31.4 (±13.2) | | 39.4 (±12.3) | 74.3 (±12.1) | |
| Water | 56.8 (±19.1) | 67.6 (±15.4) | | 60.5 (±18.6) | 82.6 (±11.5) | |

The forest disturbance detection achieved an OA of 80.5% (±3.2%) (Tables 5 and S7). The PA and UA of forest disturbance were 23.7% (±4.5%) and 78.8% (11.3%), respectively. When only forest disturbances of ≥1 ha were considered as the reference, the PA of the forest disturbance class was 36.8%. In contrast, the PA was 14.1% when disturbances of <1 ha were considered. The accuracy assessment for the LULCC/forest disturbance map revealed high classification accuracy for the stable cropland, forest, and water classes. Other classes had a lower PA and UA. In particular, the forest disturbance classes had a PA and UA of 0.3–20.7% and 9.0–50.7%, respectively (Table S3).

**Table 5.** Accuracy assessment for forest disturbance detection (1988–2019). The PA and UA for each class and the OA are shown with 95% confidence intervals.

| Class | PA (%) | UA (%) | OA (%) |
|---|---|---|---|
| Disturbance | 23.7 (±4.5) | 78.8 (±11.3) | 80.5 (±3.2) |
| No-disturbance | 98.0 (±1.1) | 80.6 (±3.4) | |

The accuracy assessment for the forest loss map of the Hansen GFC data revealed that forest disturbances (i.e., forest loss) were mapped with a PA and UA of 30.6% (±7.9%) and 69.4% (±12.1%), respectively, in the study area. The OA of the forest loss map of the Hansen GFC data (i.e., 80.4% ± 3.3%) was similar to that of the map generated in this study. The 2019 LULC map of Vietnam from Phan et al. [42] achieved a slightly higher classification accuracy (OA of 73.4% ± 3.7%) after consolidation to the LULC classes and clipping the extent in this study (Table S8). Substantial improvements were observed in the PA of the cropland and grass/shrub classes and the UA of the forest and grass/shrub classes.

### 3.2. Mapping LULCC and Forest Disturbances

The results showed that the areas of forest and settlement increased throughout the study period (area changes in Table S9). However, the areas of cropland and grass/shrub decreased. Based on the forest disturbance and corresponding LULCC results in the accuracy assessment, forest disturbances that did not result in LULCC (i.e., conversion to non-forest classes) occupied 66.8% of the total disturbance area, whereas forest disturbance that resulted in other LULC classes occupied 6.6%. The rest of the disturbances (i.e., 26.7%) were non-forest classes in 1988 but recovered to forests during the study period.

As shown in Figure 4, the mapping results indicated that cropland class was mainly distributed in lowland regions. In the mountainous regions, there was a mosaic of forest, cropland, and grass/shrub classes. In particular, the transition of abandoned cropland to grass/shrub or forests was visually ambiguous and it was difficult to correctly classify through visual interpretation. The interpreters showed the most disagreement in the initial labeling process for the cropland, grass/shrub, and forest classes.

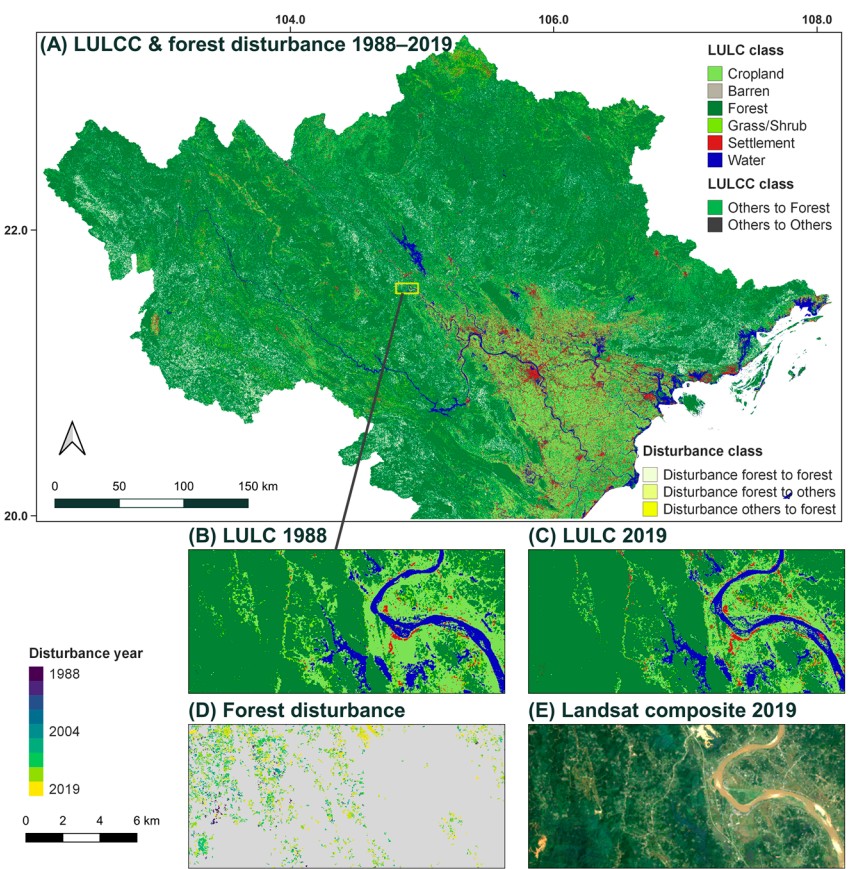

**Figure 4.** (**A**) LULCC and forest disturbance mapping of 1988–2019 for the entire study area. (**B**) Subset map of LULC 1988. (**C**) Subset map of LULC 2019. (**D**) Subset map of forest disturbances colored by disturbance years. (**E**) Subset map of 2019 Landsat RGB composite.

## 4. Discussion

LULCC and forest disturbances provide fundamental information on land surface changes and related vegetation dynamics. A temporally consistent mapping of both LULCC and forest disturbances (e.g., annual time steps) is important for informing decision-makers about the potential impacts of these changes. In this study, we examined the applicability of a LandTrendr temporal segmentation to map the annual LULCC and forest disturbances based on annual Landsat time series data in northern Vietnam. The accuracy assessment indicated relatively low accuracy for several LULC classes and disturbance detection, which we attributed to the complex landscape and environment of the study area. However, the

accuracy was comparable to that of the existing LULC/disturbance maps, indicating the importance of LandTrendr temporal segmentation combined with RF modeling. Although there is room for improvement, this study demonstrated the potential for better mapping in future research.

The accuracies obtained for LULC classification and forest disturbance detection were relatively low in this study. The PA of forest disturbances was especially lower than that of previous studies using LandTrendr in other regions (e.g., [29,92]). However, the accuracy assessment for the forest loss of the Hansen GFC data also showed lower accuracy with the same reference samples (i.e., PA of 30.6%). Thus, disturbance detection in the study area seems more challenging than in other regions. The substantial disagreement between the independent interpreters in the initial interpretation of reference samples also suggests that forest disturbances were subtle and small in the study area. Thus, it was difficult to distinguish disturbances from stable forest classes even in the visual assessment. The same difficulty was observed for spatiotemporally ambiguous LULCC in the mountainous regions, e.g., cropland to forest and grass/shrub to forests, which had similar spectral responses. The main types of forest disturbances observed in this study were temporal forest loss caused by timber harvesting, shifting cultivation, and conversion to plantation forests and permanent forest loss caused by forest conversion to non-forest classes, such as cropland. We assumed that small forest disturbances followed by rapid forest recovery hindered accurate disturbance detection because such disturbances are more difficult to detect compared with those that resulted in LULCC. Indeed, forest disturbances that remained forests, such as those caused by temporal clearing and shifting cultivation, were dominant in the study area. Small and frequent forest disturbances generated the mosaic of primary, secondary, and plantation forests, which led to a landscape of different forest types surrounded by shrub, grassland, and cropland in rural regions. Together with the heterogeneous landscape, such human-induced ephemeral forest disturbances might have made the study area complex and challenging to map.

Disturbance detection approaches that applied machine learning algorithms (e.g., RF) to Landsat time series data have increased in recent years (e.g., [29,30]) because the use of machine learning algorithms can handle complex temporal dynamics and various types of forest disturbances [93]. This study combined RF prediction with LandTrendr temporal segmentation, which has been used in previous studies (e.g., [29,94]), to both LULCC classification and forest disturbance detection. In this context, our approach is more flexible than the original LandTrendr algorithm at detecting complex changes to the land cover. However, the accuracy assessment revealed that the detection accuracy was almost the same as that of the Hansen GFC data, which is a global dataset. Although globally generated products sometimes have better or similar performance [95], most previous studies have found that locally calibrated disturbance detection achieved higher accuracies [96–98]. The results of this study can be attributed to the difficulty of characterizing forest disturbance and recovery in the study area. The detection accuracy of the LandTerndr temporal segmentation depends on the fitting results of the temporal segments. As a previous study showed that optimal parameter settings can greatly affect the accuracy of disturbance detection using Landsat time series [99], testing the parameters for the LandTrendr temporal segmentation might improve predictive performance. Although change detection algorithms that use all available Landsat observations by fitting a time series model (e.g., CCDC and BFAST) are computationally demanding, the use of such algorithms is another solution to reducing the omission error of forest disturbance detection.

This study demonstrated spatially and temporally consistent mapping of LULCC and forest disturbances. Both forest disturbances and LULCC affect vegetation dynamics and carbon sequestration. The characteristics of mapped forest disturbances that coincide with LULCC at each pixel in the time series are desirable for providing spatially explicit information. Although the LULC classification accuracy in our study was slightly lower than that of the existing LULC map for entire Vietnam [42], the forest disturbance map generated in our study has a mutually complementary relationship and thus is useful for

understanding vegetation dynamics. It was sometimes difficult to distinguish between forest, grass/shrub, and cropland in this study. The confusion can largely be attributed to the spectral similarities among these classes and ambiguous transitions that caused mixed classes in the pixels. One way to improve the classification accuracy is by adding more training data considering obtained higher accuracy for these classes in the map of Phan et al. [42]. Using indices from spectral unmixing may also solve the problem of pixels with mixed LULC classes. Because we classified LULC based on the temporal segments, the classification accuracy of LULC was also affected by the results of temporal segmentation. Thus, searching the optimal parameters of LandTrendr temporal segmentation is another possible solution for improving classification accuracy.

Among LULC classes, the largest increase in area throughout the study period was observed for the forest class (+710,556 ha, Table S9), followed by the settlement class (+390,783 ha). The observed increase in these classes between 1988 and 2019 is consistent with the findings for the entire country of Vietnam [44,100]. The decrease in area of the cropland and grass/shrub contributed to these changes, which was mainly driven by the establishment of planted forests. These transitions mainly occurred in mountainous areas where land abandonment is likely to occur [40]. Agricultural land abandonment has negative impacts on food security; however, forest regeneration can increase carbon sequestration and help generate other ecosystem services, such as timber production, biodiversity conservation, and landslide prevention.

## 5. Conclusions

In this study, we examined the applicability of LandTrendr temporal segmentation for mapping LULCC and forest disturbances based on annual Landsat time series data and RF model predictions. The results revealed that, over the three decades (1988–2019), the areas of forests and settlements in the study area (i.e., northern Vietnam) increased, whereas the areas of cropland and grass/shrub decreased. The dominance of forest disturbances that did not cause LULCC highlighted the importance of characterizing forest disturbances as well as LULCC. Although the classification accuracy was relatively low in this study because of the complex mountainous landscape and subtle forest changes, LandTrendr temporal segmentation is still useful for detecting LULCC and forest disturbances. Temporal segmentation and features derived from LandTrendr were useful for mapping both LULCC and forest disturbances. The spatially and temporally consistent mapping of LULCC and forest disturbances can provide a better understanding of landscape dynamics, which is essential for biodiversity conservation and carbon stock monitoring. Future studies should focus on improving the accuracy of forest disturbance detection, especially in areas with subtle landscape changes, as well as LULC classification in ambiguous and complex landscapes.

**Supplementary Materials:** The following supporting information can be downloaded at: https://www.mdpi.com/article/10.3390/rs15030851/s1.

**Author Contributions:** Conceptualization, K.S. and W.M.; methodology, K.S.; software, K.S.; validation, K.S., W.M. and T.F.; formal analysis, K.S.; investigation, K.S.; resources, K.S.; data curation, K.S.; writing—original draft preparation, K.S.; writing—review and editing, K.S., W.M., T.F. and R.C.E.; visualization, K.S. All authors have read and agreed to the published version of the manuscript.

**Funding:** This study was supported by the Forestry Agency of Japan.

**Data Availability Statement:** The data supporting this study will be provided by the corresponding author upon reasonable request.

**Acknowledgments:** We thank the anonymous reviewers for their valuable comments and suggestions.

**Conflicts of Interest:** The authors declare no conflict of interest.

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
