# Peer review of "Mapping Land Use/Land Cover Changes and Forest Disturbances in Vietnam Using a Landsat Temporal Segmentation Algorithm"

_remotesensing, doi:10.3390/rs15030851_

Round 1

Reviewer 1 Report

The authors can find the comments in the file herewith attached.

Reviewer 2 Report

The submission is titled "Mapping Land Use/Land Cover Changes and Forest Disturbances for Vietnam Using a Landsat Temporal Segmentation Algorithm " and it is about using satellite data to classify land use and land cover in Vietnam. Authors state that this work is relevant and timely because accurately mapping land use/land cover changes (LULCC) and forest disturbances provides valuable information for understanding the influence of anthropogenic activities on the environment at regional and global scales. Many approaches using satellite remote sensing data have been proposed for characterizing these long-term changes. However, a spatially and temporally consistent mapping of both LULCC and forest disturbances is still lacking despite their critical contributions to the carbon cycle. In this study, we examined the applicability of Landsat time series temporal segmentation and random forest classifiers to mapping LULCC and forest disturbances in Vietnam. We used the temporal segmentation algorithm to derive key features of land use/land cover transitions and forest disturbances from annual Landsat time series data. We developed separate random forest models for classifying land use/land cover and detecting forest disturbances at each segment and then derived LULCC and forest disturbances during the period of 1988–2019. The results showed that both LULCC classification and disturbance detection achieved low accuracy in several classes; however, the level of accuracy was comparable to that of existing datasets using the same reference samples in the study area. We found relatively high confusion between several land use/land cover classes (e.g., grass/shrub, forest, and cropland) that can explain the lower classification accuracy. The mapping of forest disturbances suggested that most forest disturbances were followed by forest recovery, not by transitions to other land use/land cover classes. The landscape complexity and ephemeral forest disturbances contributed to the lower classification and detection accuracies in this study area. Using more training samples and effective variables would potentially improve the classification and detection accuracies. Nevertheless, our approach was shown to consistently map both LULCC and forest disturbances, and it can potentially contribute to better mapping of both long-term changes.

In general, the paper is well written. The language is clear and discussions are well-presented. However, there is hardly any novel aspect to the presented work. Using satellite images to classify not only land cover but also buildings etc, is a well investigated topic. Similar work has been done in different countries for different purposes such as agriculture, water resource management,  power system planning, etc.. In almost all of these cases the results have high accuracy unlike the work presented in this manuscript.

The only way in which this work may be considered for publication is if it is presented as a negative result and directions for further enhancement are recommended. In other words, if the proposed approach has same low accuracy levels as other approaches presented in the literature, what is the contribution to the current body of knowledge? What is the novelty presented herein?

These questions are not answered by the authors anywhere in the manuscript, not at the end of the introduction where it is traditional to highlight novelty of the presented work, nor in conclusions where important takeaways are presented.

Literature review is not bad, but the number of references is too high. The reason why this presents itself as a problem is as follows: The references are not exclusive to land use classification domain. There are many works in the literature where satellite images are utilized for different purposes. Authors need to focus citing these papers (rather than overall 80+ references) and highlight why/how their approach is relevant and valuable. Considering that the proposed scheme is not simpler nor more accurate, there is no real motivation to adopt this scheme or consider it for academic or practical purposes.

In short, authors need to revisit the literature on the use of satellite images for land classification. They need to focus on the ones with high accuracy and compare their approach with them. There needs to be a motivation for the presented work. Is it a certain technology/ or time period or location? (There are many studies focusing on classification of vegetation in Vietnam, though)

Reviewer 3 Report

In the Manuscript entitled “Mapping Land Use/Land Cover Changes and Forest Disturbances for Vietnam Using a Landsat Temporal Segmentation Algorithm”, authors performed detailed examination to know the applicability of LandTrendr temporal segmentation for mapping LULCC and forest disturbances based on annual Landsat time series data and RF model predictions. I have this study technically sound and well presented. However, I have a few suggestions as follows:

·       Elaborate the terms forest disturbance, stable forest and buffer forest.

·       Justification is required for images collected between 1st of May till 30th of November, how it's helps in reduction of yearly variation caused by seasonal changes.

·       Figure 3 can be improved because the text inside the graph is clumsy.

·       Please mention the reference for all equations.

·       Even though study is achieving all the objectives, accuracy over classification can be improved inspite of the complication in study area.

·       Study emphasizes more towards forest disturbance. Therefore, there is a need to discuss the kinds of forest disturbances observed.

L 151: Abbreviate the terms

L 220-221: Is it necessary to ignore possible multiple LULCC? if yes justify

L 226 and L355: Please mention the correct Table numbers.

Reviewer 4 Report

Review of “Mapping Land Use/Land Cover Changes and Forest Disturbances for Vietnam Using a Landsat Temporal Segmentation Algorithm”

This manuscript describes the process of generating map products of Land Use / Land Cover (LULC), LULC Change, and forest disturbance across northern Vietnam. The study area displays variation in topography, forest types, and land use. The underlying data were Landsat images collected between 1987 and 2021, which were processed using the LandTrendr algorithm. LandTrendr segments were the basis of the modeling effort which assigned a LULC category to each LandTrendr segment. The final maps were validated using 792 reference points, which were chosen following a stratified random design and were used to calculate accuracy and areal estimates of LULC, and LULC Change, and forest disturbance. The authors further compared their results to two existing maps: the Hansen Global Forest Change (a global product) and Phan (a national product) data products.

This study was well described, well written, and used established analytical techniques.

The authors followed Olofsson’s “good practices” recommendations including: selecting appropriate number of reference observation, including confidence intervals in accuracy assessments, and reporting complete error matrices (proportions and map class weights).  

There are 2 items where I would prefer clarification.

First, perhaps I am mistaken, but it appears that non-forest area was used when calculating the accuracies in Table 5. If this is true, then I believe the table should have 3 map classes: disturbed forest, not disturbed forest, non-forest. In my opinion, it does not seem appropriate to combine non-disturbed forest with the non-forest classes (some of which experienced “other-to-other” changes). Or if the intent is to report the forest disturbance accuracies, then I believe it would be more appropriate to remove the non-forest area (limit the scope of inference for Table 5 to just forested land). The same would apply for Table S6.

Second, I am unclear how the author’s defined agreement when analyzing the disturbance patch size (L 286-290). The manuscript states disturbances were manually delineated. Were both the disturbance maps and the Landsat RBG images delineated? Was agreement defined if the reference pixel intersected both delineated polygons? Or did a specific proportion of the two polygons have to intersect before agreement was recorded? Additional information would help my understanding.

Minor comments
L 20: (approx.) - Please mention in the abstract that you are using LandTrendr.
L 42: “replanting” instead of replantation.
L 47: The word “there” is not needed.
L180 – 181: “The calculation details of the predictor variables are summarized in Table S2.” Table S2 describes the predictor variables, but does not provide details regarding the calculations. Please update this sentence to reflect this fact.
L 276: “wheter” -> “whether”
L 284: “lables” -> “labels”
L 341 – 343: I’m confused here. Is the OA of 73.4% for the complete Phan map, or is that the OA calculated using the validation points you interpreted and the portion of the Phan map that intersected your study area?

Table S6. Please state in the caption that these results are for forest land.

Round 2

Reviewer 1 Report

Comments to authors:

1 - Referring to my last report, this article is very weak in terms of mapping (for example in the flowchart (Figure 2) the authors mentioned NDVI, NDMI, etc.., for that we need to see maps each period from 1988 until 2019).

2 - From the last version to the current version, the reviewer not found deeply correction especially Remote Sensing and GIS mapping and simulations.

Author Response

Please see the reply to the academic editor.

Reviewer 2 Report

Authors did not take the reviewer comments into account. Virtually there is no revision done in this submission. This includes "not highlighting novel aspects of this work at the end of introduction" which is almost agreed upon in journal papers. If the authors cannot reflect such a simple revision on their manuscript, the revision will not be satisfactory.

Author Response

(The authors gave the same response as above.)
